# Risk Transmission of Trade Price Fluctuations from a Nickel Chain Perspective: Based on Systematic Risk Entropy and Granger Causality Networks

**DOI:** 10.3390/e24091221

**Published:** 2022-08-31

**Authors:** Xuanru Zhou, Shuxian Zheng, Hua Zhang, Qunyi Liu, Wanli Xing, Xiaotong Li, Yawen Han, Pei Zhao

**Affiliations:** 1Institute of Mineral Resources, Chinese Academy of Geological Sciences, Beijing 100037, China; 2School of Economics and Management, China University of Geosciences, Beijing 100083, China; 3School of Management, China University of Mining and Technology, Beijing 100083, China; 4Development Research Center of China Geological Survey, Beijing 100037, China; 5School of Economics and Management, Zhejiang Sci-Tech University, Hangzhou 310018, China; 6Teacher’s College, Beijing Union University, No. 5 Waiguan Oblique Street, Chaoyang District, Beijing 100011, China

**Keywords:** nickel industry chain, trade prices, granger causality networks, systemic risk entropy

## Abstract

Nickel is a strategic mineral resource, with 65% of nickel being used in stainless steel. The situation in Ukraine starting in February 2022 has led to significant fluctuations in nickel prices, with prices of nickel products along the same chain affecting and passing through each other. Using systematic risk entropy and granger causality networks, we measure the volatility risk of trade prices of nickel products using the nickel industry chain trade data from 2000–2019 and explore the transmission patterns of different volatility risk prices from the industry chain perspective. The findings show that: (1) Nickel ore has the highest risk of import trade price volatility and a strong influence, but low risk transmission. Stainless steel has the highest trade price impact but is also subject to the strongest passive influence. (2) The Americas have a higher risk of trade price volatility but a weaker influence. The influence and sensitivity of trade prices is stronger in Asia and Europe. (3) Indonesia’s stainless steel export prices have a high rate of transmission and strong influence. Germany’s ferronickel export prices are highly susceptible to external influences and can continue to spread loudly. Russian nickel ore export prices are able to quickly spread their impact to other regions.

## 1. Introduction

Nickel is mainly used to make stainless steel and other high-temperature and corrosion-resistant alloys. In addition, nickel also plays a crucial role in power generation, aerospace, money manufacturing, and military fields [1,2]. The most important battery type in electric vehicles is the lithium-nickel-manganese-cobalt oxide (NMC). The cathode of these lithium-ion batteries consists of 30–72% nickel [3]. Therefore, nickel will be the most promising metal-for-battery production [4]. There is strong spatial variability in global nickel supply and demand, and the U.S. has included nickel as a key mineral in February 2022. In 2020, the global nickel ore reserves were about 94 million tons, mainly concentrated in Indonesia (22%), Australia (21%), and Brazil (17%). Actual nickel production was concentrated in Indonesia (34%) and the Philippines (16%). As the largest consumer of nickel, China accounts for only about 4% of global production [5]. International trade is an important way to solve the uneven spatial distribution of nickel resources [6], so it is important to study the nickel trade market to ensure the security of nickel resource supply and the maintenance of a good trade environment.

Often international trade markets are not always stable, which is closely related to factors such as national politics, regional disputes, global financial stability, and resource demand. Market instability can lead to dramatic fluctuations in commodity prices. For example, in March 2022, the price of nickel on the London Stock Exchange fluctuated greatly, with the continuous rise and fall limits, while the change of nickel spot price was closely related to the futures price [7]. The Russian nickel exports were restricted after the Russian-Ukrainian conflict. Meanwhile, the growth of downstream demand for new energy vehicles has led to a further increase in nickel demand [8]. Russia is an important supplier of nickel resources in Europe, and European nickel is also facing supply risks due to trade sanctions. A variety of factors together have led to dramatic fluctuations in international nickel trade prices. Therefore, it is important to assess the risk of nickel trade price fluctuations to ensure the stability of nickel supply and demand in each country.

Entropy is an important indicator of system stability, and researchers have used entropy to measure the volatility risk of prices: Xiao et al. (2018) introduced systematic risk entropy to analyze regional differences and synchronization of the Chinese consumer price index [9]; Xiao (2021) defined systematic risk entropy, synchronization ratio, and stability based on inter-provincial retail price index data in China to quantitatively analyze the market evolution characteristics [10]. These studies provide a valuable reference for measuring the volatility risk of prices but do not consider the transmission of volatility risk. Price differentials between products are a prerequisite for the transmission of price fluctuations [11]. The same product may belong to different subjects or regions, and if there is a price difference between subjects, the product will be transferred between subjects and the product exporter may increase the trade price. As the inflow of the product changes the market supply and demand, the trade price of the buyer of the product is in turn restricted, which leads to price transmission between different agents [12,13]. Subjects with higher price volatility risk are more likely to pass on risk to other subjects or regions. For different types of products, the transmission of volatility risk between prices remains. Influenced by factors such as resource competition and product utility substitution, price differences can cause producers to increase the market supply of higher-margin products, which leads to the transmission of price volatility across products. In addition, when different products are part of the same supply chain, that is, one product is a raw material or finished product produced using another product, fluctuations in raw material prices will inevitably raise the production costs of the product, causing prices to pass between products. Products with a higher risk of price fluctuations are also more likely to pass on the risk to other products.

Granger causality tests are often used to study information transmission and causality between prices [14,15], but due to the large number of countries involved in international trade, Granger causality tests are sometimes difficult to take all the subjects into account. Complex networks are a suitable approach for multi-subject modeling and have been widely used in the study of metal prices [16,17]. Therefore, researchers have combined Granger causality tests and complex network theory to study the return spillover structure, price transmission relationships, and network structure changes before and after unexpected events in the stock market from multiple perspectives in the time and frequency domains [18,19,20,21,22,23]. Not only that, but the Granger causal networks are also often used to study the transmission relationships among price indices [24,25]. While these studies discuss inter-price transmission relationships from a systematic perspective, they do not take into account the industry or chain hierarchy in which the product is located.

Based on the nickel processing and production process, the nickel product chain can be divided into upstream ore mining, midstream smelting, and downstream product processing. Each stage produces different types of nickel products, and the products of the previous stage are often the raw materials for the next stage of production. Among the chain, the prices of products at different stages and at the same stage affect each other [26,27,28]. In summary, the transmission relationship of the price of nickel products needs to be studied from the perspective of the industrial chain. Pyrometallurgy is the most common smelting method of nickel ore, and its product, ferronickel, is the most important raw material for the production of stainless steel [29,30]. About 65% of global nickel is used to make stainless steel [31,32], not only that, but as a precious metal, nickel’s price impact on the cost stability of stainless steel is the most important factor limiting the use of stainless steel [33]. Therefore, we choose nickel ore, ferronickel, and stainless steel as representative products in the upstream, midstream, and downstream of the nickel industry chain to study the risk transmission of price volatility among different links of the nickel industry chain.

In recent years, the transmission pattern of price fluctuations among industrial chains has become the focus of scholars’ attention. The current relevant studies mainly focus on the stock prices of the real estate industry [34] and steel product prices [12,13], and pay attention to the diffusion path and transmission pattern of prices. These studies provide valuable references to explore the volatility risk transmission of prices from the perspective of industry chains, but there are still shortcomings. First of all, most of the studies only consider part of the chain’s hierarchy and are more often conducted on a single market. In addition, these studies only consider the price transmission effect and do not include the possibility of price volatility. Finally, most of these networks are unpowered and cannot quantify price impact.

In summary, to measure the volatility risk of trade prices of nickel products and explore the volatility transmission law of trade prices from the perspective of the industry chain, this study calculated the systematic risk entropy of trade prices of different regions and products and constructed a Granger causality network of trade prices based on nickel industry chain. The main contributions of the work are as follows: (1) Based on the perspective of the nickel industry chain, the systematic risk entropy of nickel product prices is calculated from both product types and geographical regions to measure the volatility risk of trade prices. (2) A weighted directional Granger causality network is constructed, and the network is split according to the transmission relationship between different products and different regions to deeply characterize the transmission characteristics of trade prices of different products and different regions. (3) The trade prices of the top 10 influential nickel products are screened and their volatility transmission characteristics are discussed in six dimensions. The study helps policymakers to measure the risk of trade price volatility of different products and supports the Formulation of trade policies.

The paper is organized as follows: Section 2 describes the methods and data used in the study. Section 3 gives the results and the corresponding discussion. The final section contains conclusions and policy recommendations.

## 2. Data and Method

### 2.1. Data

The study used trade data for nickel ore, ferronickel, and stainless steel from 2000 to 2019. All data used in the study came from UN Comtrade (https://comtrade.un.org/, accessed on 5 May 2022). At the time of data acquisition, the HS code was assigned to each trade commodity. HS codes for nickel ore and ferronickel are 2604 and 720260. At the time of statistics, stainless steel is divided into six types, their HS codes are 7218, 7219, 7220, 7221, 7222, and 7223. Therefore, the stainless steel trade data used in the study is the sum of six kinds of stainless steel trade data. The price of the trade is calculated by dividing the volume of trade by the value of trade. Most countries have not been involved in nickel products for 20 consecutive years, leading to a large number of samples missing. To ensure the stability and typicality of data selection, countries ranking 10 in the import or export of three nickel products were selected as samples (Table A1). These countries account for more than 60% of the total trade volume and are sufficiently representative (Table A2).

### 2.2. Construction of Weighted and Directed Granger Causal Networks

The research purpose of this work is to explore the pass-through relationship between the prices of different types of nickel products in different countries. We use Granger causality to express the pass-through relationship between prices. The reason is that Granger causality can highlight the chronological order in which two variables occur. The product trade prices of countries in the past 20 years constitute multiple time series (which are the objects of the Granger causality test). The results of Granger causality test will be the premise of constructing Granger causality network, which takes time series as nodes and Granger causality between nodes as edges. The following is a detailed description of the model construction process.

(1)Unit root test

The Granger causality test requires that the time series be stationary, otherwise false regression may occur (the reason is that the lag coefficient in the autoregression process is 1, so any error of the residual sequence will not attenuate with the increase of sample size). The unit root test can be used to test the stationarity of time series. In this study, the Augmented Dickey-Fuller test (ADF test) was used to determine whether the unit root existed in time series. The test formula is as follows, where, $y_{t}$ is the y value in the t period (y represents the time series of national nickel product trade prices in 20 years), ${\Delta y}_{t-i}$ is the lag difference term, and $\varepsilon_{t}$ is the disturbance term. After examination, it is found that some sequences are not stationary. We perform first-order difference processing on all-time series and obtain all stationary sample data (Table A3).
(1)$$y_{t}=\beta_{0}+\rho y_{t-1}+\overset{p-1}{\underset{i=1}{\sum }}\gamma_{i}\Delta y_{t-i}+\gamma t+\varepsilon_{t}$$

(2)Co-integration test and Granger causality test

If the linear combination of multiple non-stationary time series with the same order is stable, there is a cointegration relationship among these time series. Var model is constructed for the co-integration test to determine whether there is a long-term equilibrium relationship between sequences. There are two commonly used co-integration test methods: Engel-Granger two-step test and Johansen co-integration test. The Johansen test adopts multiple equation technology, which is less limited. Therefore, the Johansen-Juselius test method is used to test the co-integration relationship between time series [35,36].

If there is a long-term equilibrium relationship between prices, we can continue to calculate the Granger causality between them. After testing the co-integration relationship between prices, the granger causality test is carried out on the price combinations that pass the co-integration test. The equation of the Var model is as follows, where $y_{t}^{1}$ and $y_{t}^{2}$ respectively represent the value of two different trade variables in t period, $y_{t-i}^{1}$ and $y_{t-i}^{2}$ are their lag terms, $\varepsilon_{t}^{1}$ and $\varepsilon_{t}^{2}$ are the random perturbation terms of the equation, $a_{i}^{1}$ and $a_{i}^{2}$ are the Granger coefficients, and m and n are the maximum lag order.
(2)$$y_{t}^{1}=\overset{m}{\underset{i=1}{\sum }}a_{i}^{1}y_{t-i}^{1}+\varepsilon_{t}^{1}$$
(3)$$y_{t}^{2}=\overset{n}{\underset{j=1}{\sum }}a_{i}^{2}y_{t-i}^{2}+\varepsilon_{t}^{2}$$
(4)$$y_{t}^{1}=\overset{m}{\underset{i=1}{\sum }}b_{i}^{1}y_{t-i}^{1}+\overset{m}{\underset{j=1}{\sum }}c_{i}^{1}y_{t-i}^{2}+\xi_{t}^{1}$$
(5)$$y_{t}^{2}=\overset{n}{\underset{j=1}{\sum }}b_{i}^{2}y_{t-i}^{2}+\overset{n}{\underset{i=1}{\sum }}c_{i}^{2}y_{t-i}^{1}+\xi_{t}^{2}$$

To test the Granger causality between variables, we introduce lag variables between variables into Formulas (2) and (3), and get Formulas (4) and (5). The meaning of each variable is the same as Formulas (2) and (3). In Formula (4) and (5), the null hypothesis is that both $c_{i}^{1}$ and $c_{i}^{2}$ are 0, that is, the introduced variable will not improve the prediction effect of the original variable. If the null hypothesis is rejected, it can be proved that the introduced variable is the Granger cause of the original variable. We set a significance level of 0.05.

(3)Lagged cross-correlation coefficient

The Granger causality between prices cannot reveal the intensity of their causality. The Pearson correlation coefficient is widely used to calculate price correlation [37]. In fact, the correlation between prices involves a certain lag. Pearson’s correlation underestimates the correlation between prices that include lag. The lag correlation number is used as the Granger causality intensity between prices [25]. the calculation formula is below:(6)$$LC_{ij}=\frac{\sum_{t}\left[\left(i_{t}-mean\left(i\right)\right)\times \left(j_{t+d}-mean\left(j\right)\right)\right]}{\sqrt{\sum_{t}{\left(i_{t}-mean\left(i\right)\right)}^{2}}\sqrt{\sum_{t}{\left(j_{t}-mean\left(j\right)\right)}^{2}}}$$
where mean(i) and mean(j) are the mean values of price i and price j respectively. $i_{t}$ and $j_{t}$ are the prices i and j at time t. d is the lagging period. The maximum value of d is set as 5, and the maximum value of ${\mathrm{LC}}_{\mathrm{ij}}$ is selected as the granger causality intensity between prices.

(4)Network construction

In this study, ${\mathrm{GC}}_{\mathrm{ij}}$ was used to represent the conduction relationship between nickel product trade prices i and j. When price i is the Granger cause of price j, ${\mathrm{GC}}_{\mathrm{ij}}$ is 1, that is, there exists an edge from node i to node j; When price j is the Granger cause of price i, ${\mathrm{GC}}_{\mathrm{ji}}$ is 1, that is, there exists an edge from node j to node i. In contrast, when there is no Granger causality between the two nodes, ${\mathrm{GC}}_{\mathrm{ij}}$ or ${\mathrm{GC}}_{\mathrm{ji}}$ is 0 [34]. A weighted directed Granger causality network is established with trade price as a node, Granger causality between prices as edge, and lag correlation between prices as weight (WDGCN). The network is represented by Formula (7) and Figure 1.
(7)$$WDGCN=\left[\begin{array}{ccc} 0 & \cdots & DGC_{1N}\\ \vdots & \ddots & \vdots \\ DGC_{N1} & \cdots & 0 \end{array}\right]$$
where ${\mathrm{DGC}}_{\mathrm{ij}}$ is the weighted Granger causality between price i and price j. ${\mathrm{DGC}}_{\mathrm{ij}}{\;\mathrm{is}\;\mathrm{obtained}\;\mathrm{by}\;\mathrm{multiplying}\;\mathrm{GC}}_{\mathrm{ij}}\;\mathrm{and}\;\left|{\mathrm{LC}}_{\mathrm{ij}}\right|.$

### 2.3. Trade Transmission Network Analysis

The complex network integrates trade prices and the transmission relationship between them and systematically presents the transmission characteristics of trade prices. Six network indicators are used to describe the transmission characteristics of prices.

(1)Impact

In a network, the weighted out-degree represents the sum of the weights of the edges of the nodes. In the price transmission network, the weighted out-degree of nodes represents the sum of weighted Granger causes of other prices. Then, we believe that the weighted out-degree represents the influence level of the price, which is expressed by Formula (8):(8)$$S_{i}^{out}=\overset{N}{\underset{j=1}{\sum }}DGC_{ij}$$
where ${\mathrm{DGC}}_{\mathrm{ij}}$ is the weighted Granger causality between price i and price j.

(2)Extent of being affected

In a network, the weighted in-degree represents the sum of the weights of the edges pointing to nodes. In the price transmission network, the weighted in-degree of nodes represents the sum of weighted Granger results of other prices. Then, we believe that the weighted in-degree represents the degree to the price being affected, which is expressed by Formula (9):(9)$$S_{i}^{in}=\overset{N}{\underset{j=1}{\sum }}DGC_{ji}$$
where ${\mathrm{DGC}}_{\mathrm{ji}}$ is the weighted Granger causality between price j and price i.

(3)Transmission range

In a network, the out-degree is the number of edges that indicate a node. In the price transmission network, the price out-degree describes how much trade price will be affected by the trade price fluctuation [24]. We believe that the out-degree represents the transmission range of price, which is expressed by Formula (10):(10)$$k_{i}^{out}=\overset{N}{\underset{j=1}{\sum }}GC_{ij}$$
where ${\mathrm{GC}}_{\mathrm{ij}}$ is the Granger causality between price i and price j.

(4)Sensitivity degree

In a network, in-degree represent the number of edges pointing to nodes. In the price transmission network, the in-degree of a price describes how much the trade price will be affected by other trade prices [24]. We believe that the in-degree represents the range affected by the price (i.e., price sensitivity), which is expressed by Formula (11):(11)$$k_{i}^{in}=\overset{N}{\underset{j=1}{\sum }}GC_{ji}$$
where ${\mathrm{GC}}_{\mathrm{ji}}$ is the Granger causality between price j and price i.

(5)Transmission speed

In a network, closeness centrality measures the average length of the shortest path from each node to the others. In the price transmission network, the greater the closeness centrality of the trade price is, the shorter the transmission path exists between the trade price and other trade prices, that is, the faster the fluctuation characteristics of the price can be transmitted between them [12]. Then we believe closeness centrality represents the transmission speed of price fluctuations. The closeness centrality is expressed by Formula (12).
(12)$$CC_{i}=\frac{N-1}{\sum_{j=1}^{N}d_{ij}}$$
where $d_{\mathrm{ij}}$ is the shortest conduction path length from price i to price j, and N is the quantity of trade price in the system.

(6)Transmission hubs

Betweenness centrality represents the number of times a node acts as the shortest bridge between two other nodes. In the price transmission network, the stronger the betweenness centrality of trade price is, the more it acts as the intermediary of causality among other trade prices [12]. Therefore, we use betweenness centrality to measure the transmission centrality of price, that is, the degree of influence of price on transmission among other prices.
(13)$$BC_{i}=\underset{s\neq i\neq t}{\sum }\frac{n_{st}^{i}}{g_{st}}$$
where $n_{\mathrm{st}}^{i}$ is the number of times that price i passes through the conduction path between the other two prices, and $g_{\mathrm{st}}$ is the number of the shortest circuit in the system.

### 2.4. Systemic Risk Entropy

The study needs to introduce indicators to measure the volatility risk of the trade prices of nickel products in different products and regions. There is evidence that the eigenvalue and eigenvector of the correlation matrix have a good effect on measuring system risk [38,39]. Some scholars used random matrix theory to study the systemic risk of the American real estate market and found that the feature vector of the correlation matrix either contains geographic information, the different degree of housing price growth rate, or both [40]. In addition, Meng et al. (2014) proposed to use $\sum_{i=1}^{H}\lambda_{i}/N$ to measure systemic risk, because this index perfectly integrates the weak correlation of market performance [40]. Xiao et al. (2018) believes that $\sum_{i=1}^{H}\lambda_{i}/N$ is only the sum of the eigenvalues, which ignores the extent to which the eigenvalues reflect real market information. Xiao et al. (2018) improved Shannon’s entropy and used systematic risk entropy to more reasonably describe the systemic risk of prices [25]. The calculation Formula of system risk entropy is as follows [9,25]: (14)$$SRE=\left(-1/log\left(N\right)\right)\overset{H}{\underset{i=1}{\sum }}\left(\lambda_{i}/N\right)log\left(\lambda_{i}/N\right)$$
where $\lambda_{i}$($\lambda_{i}>0$) is the eigenvalue of matrix WDGCN, H is the number of $\lambda_{i},i=1,2,\cdots ,H$, and N is the number of trade prices in the system. H≤N and 0≤SRE≤1.

## 3. Results

### 3.1. Systematic Risk of Price Fluctuation

The trade prices of nickel ore, ferronickel, and stainless steel show similar volatility characteristics (Figure 2). From 2000, the price of nickel products showed a continuous fluctuation upward, reaching a peak in 2007. The year 2008, affected by the global financial crisis, stainless steel manufacturing industry downturn, the price of raw materials and stainless-steel products are showing a precipitous decline. From 2009 to 2012, the global economy rebounded, industrial capacity recovered, downstream product demand was able to expand, and product prices increased. From 2012 to 2019, nickel product prices gradually stabilized.

While the overall trend in trade price volatility is broadly similar, there are still differences in trade prices between countries. Canada’s nickel ore import prices maintain a high level, while there is a more pronounced volatile variation (unit prices fluctuate between $5 and $35) Nickel ore export prices in the Philippines and the U.S. in 2014, and in Brazil and the U.S. in 2019 are significantly higher, which may be related to Indonesia’s mining ban policy [41] (Indonesia introduced a ban on nickel ore in 2014 and announced a total ban on nickel in 2019 ore exports). The import prices of nickel and iron in the United States and India remained high and volatile. Inventories in the U.S. stainless steel industry have remained low and its stainless steel supply is having difficulty meeting market demand [42]. In addition, the severe shortage of materials in the U.S. market has contributed to the increase in ferronickel and stainless steel prices. The lack of container supply and high transportation costs have further increased the import prices of ferronickel in the United States [43]. India overtook Japan as the world’s second-largest producer of stainless steel in 2016, after China [44]. Ferronickel imports are an important support for India’s stainless steel production. South Korea and the Republic of Dominica have higher prices for ferronickel exports. Although China is the world’s largest producer of stainless steel, its relatively low export prices reflect, to some extent, the possible low value-added problem of stainless steel production in China.

Nickel products are subject to large fluctuations in trade prices, making it necessary to study their volatility risk. The magnitude of fluctuations in a single price is easier to observe, but fluctuations in multiple price curves interfere with the judgment of trends and do not allow for quantitative comparisons. Therefore, the volatility risk of price systems (multi-product price composition price systems) is difficult to measure. Systematic risk entropy can measure the volatility risk of a price system. We quantify the volatility risk of trade prices from both product and regional perspectives (Table 1 and Table 2). It should be noted that the closer the entropy of systematic risk is to 1, the lower the risk of trade price volatility is, and the opposite is true.

The calculated values of systematic risk entropy are concentrated between 0.8 and 0.95, indicating that the overall volatility risk of trade prices is low. This may be related to the time scale of the data, with some researchers demonstrating that high-frequency data can respond to more volatility information [45]. Due to the limitation of data availability, the study used annual trade prices for the calculation of risk entropy. Although the level of risk entropy may be overestimated, it still enables the comparison of volatility risk across different types and regions of trade prices. In addition, we assign regional attributes to each price to compare the volatility risk of trade prices across regions (Table A1).

Nickel ore trade prices have the highest risk of volatility, with import prices significantly more volatile than trade prices of other nickel products (Table 1). Nickel ore is at the forefront of the industry chain, playing a pivotal role in stainless steel production, the electroplating industry, and new energy battery manufacturing. The supply of raw materials, on the other hand, is mainly controlled by resource countries, whose corresponding policy uncertainties are higher. For example, countries such as Indonesia, Australia and Brazil hold 60% of the global nickel resources [46]. To develop high-tech industries and try to turn resource advantages into economic advantages, Indonesia has repeatedly enacted policies to ban the export of nickel ore, which makes the trade price of the nickel ore market more unstable [47]. Stainless steel is downstream of the industry chain, and its price is more directly influenced by the end-consumer demand, so it is more stable compared to nickel ore price. There is a high risk of volatility in trade prices in the Americas (Table 2), which may be related to the trade protection policies it pursues [48]. The systematic risk entropy of trade prices in the rest of the regions is above 0.9, and the risk of price volatility is low.

### 3.2. Granger Causality Network

In Section 3.1, we find that there are similar fluctuation characteristics among the trade prices of different products, which indicates that the fluctuation risk is transmitted among the trade prices. To study the transfer law of trade prices, we build a weighted and directed Granger causality network and use the thermodynamic diagram to represent the conduction coefficient between prices (Figure 3). The main color of the heat map is dark blue, indicating that there is no Granger causality between most of the prices. While most of the colors beyond dark blue are light yellow, indicating that most of the conductivity coefficients between prices are above 0.8 and have a strong correlation. There is no clear pattern in the distribution of Granger causes, while most of the Granger results focus on time series 38 to 60, these time series correspond to stainless steel prices in Europe and Asia (Table A1). This suggests that stainless steel prices in Europe and Asia are more susceptible to the trade prices of other products in the chain.

The study uses six indicators to measure the transmission characteristics of trade prices. WID, WOD, ID, OD, CC, and BC denote the influence, degree of being influenced, range of influence, sensitivity, speed of transmission, and pivotality of transmission of trade prices, respectively. Figure 4 shows the correlations among the indicators and the distribution characteristics of trade prices. Since there is a significant positive correlation between WID, WOD, ID, and OD, only the correlations among WID, WOD, ID, and OD are studied in this section.

There is a positive correlation between WID and WOD (Figure 4), which indicates that the more influential trade prices are also more influenced by other prices. there is no significant correlation between CC and WID and WOD, indicating that the speed of transmission is not related to the degree of influence and being influenced. there is a positive correlation between BC and WID and WOD, indicating that the more influential trade prices are more likely to be the transmission core. In addition, the transmission characteristics of most trade prices are small influence, low degree of influence, slow transmission speed, and weak transmission hub, which indicates that the Granger causality network composed of trade prices has obvious scale-free characteristics [49].

### 3.3. Risk Transmission among Different Nickel Product Prices

The study calculates the transmission indicators for different products. The product’s transmission indicator is obtained by averaging the indicator values of all trade prices under that product. In addition, calculations are done from both import and export perspectives (Table 3). The trade price of stainless steel has a higher influence and is also more susceptible to the other trade prices, and its transmission speed is also better than the other two nickel products. However, the pivotal nature of stainless steel trade price transmission is poor, which means that the stainless steel trade price has less influence on the transmission between other prices. Stainless steel export prices have the strongest influence, while import prices have the highest sensitivity. This indicates that the bargaining power of stainless steel exporters is stronger.

The influence of ferronickel trade prices is between that of stainless steel and nickel ore. Ferronickel export prices possess a stronger sensitivity, indicating that ferronickel export prices are more susceptible to the influence of other trade prices. This may be due to the fact that ferronickel, as an intermediate link in the chain, is more susceptible to the dual impact of upstream and downstream market fluctuations in its trade market. Although the volatility risk of nickel ore trade price is higher, the transmission ability is not strong, which means that the price volatility risk is not easily transmitted to other links of the industry chain. Nickel ore has a stronger transmission pivot, which means that nickel ore trade prices have a stronger influence on the transmission between other prices, so the higher volatility risk of nickel ore trade prices still needs to be watched.

The results of Table 3 are concerned with the conduction patterns of the overall network. Granger causality within a product or between two different products is constructed as multiple independent Granger causal networks, and these networks are used to study the conduction laws between products (Figure 5 and Table 4).

The Granger causality between the trade prices of nickel ore and stainless steel and the Granger causality between the trade prices of ferronickel and stainless steel form a tighter transmission network that has more nodes, edges, and a shorter average distance (Table 3). This shows that the upstream and midstream products in the chain are more likely to have a price transmission relationship with the downstream products. The longest transmission path between nickel ore trade prices and between nickel ore trade prices and ferronickel trade prices is 5, indicating a longer transmission chain between upstream prices and downstream prices, which is more likely to have a multi-level impact.

### 3.4. Risk Transmission among Different Regions

Different regions also have different patterns of price transmission. The study examines the transmission characteristics of price fluctuations from a regional perspective by regionally classifying the major countries involved in the trade of nickel products (Table 5 and Table 6). Since only South Africa is classified as Africa in the scope of the study, it is not discussed separately. There is a strong influence and sensitivity of the trade price of nickel products in Asia, and the influence of trade price in Asia mainly comes from the price of stainless steel (Table 5). in 2020, Asia ranks first in the world in stainless steel production with a 72% share [44]. Global stainless steel production growth will be dominated by Asia due to its huge consumer market and industrial upgrading. Asia’s stainless steel trading partners are all over the world, and their price changes will inevitably have an impact on the global nickel industry chain trade prices.

Europe’s trade prices are second only to Asia in terms of influence and sensitivity. In addition, Europe has a stronger price transmission speed. Unlike Asia, Europe’s trade price influence is concentrated in the upstream and midstream of the industry chain. Europe itself lacks nickel resources, and most of its nickel consumption relies on imports. The European economy relies mainly on high value added from the deep processing of primary products. For example, nickel-iron alloys produced in China and India contain about 10% nickel, while the nickel content of European nickel-iron alloys is usually in the range of 20–30% [50]. The high nickel content of ferronickel is also becoming a raw material for new energy battery manufacturing, and with the development of the new energy vehicle industry, the influence of nickel refining and processing in Europe will be further increased [51].

Compared to Asia and Europe, the ability of trade price transmission in the Americas and Oceania is not outstanding. The most influential trade prices in the Americas and Oceania are concentrated upstream of the chain (Table 6). The Americas and Oceania are rich in nickel resources, with countries such as Australia, New Caledonia, Colombia, and Brazil accounting for more than 50% of global nickel reserves.

Granger causality within regions or between two different regions was constructed as multiple independent Granger causal networks, and these networks were used to study the conduction patterns between regions (Figure 6 and Table 7). The study screened the networks using a threshold of 10 edges, i.e., regional transmission networks with less than 10 edges were not considered.

The Granger causality between Asian and European trade prices forms a much larger transmission network with more connected edges (Table 7). This indicates that Asia is more likely to have an inter-price transmission with Europe. However, the network tightness of the Asia-Europe trade price transmission network is not high, with an average transmission distance of 3.14 and the longest transmission path of 9, which indicates that there is a longer transmission chain between Asia and Europe. In addition, intra-European and intra-Asian trade prices also have more transmission chains, which means that when prices change in two regions, it will be easier for intra- and inter-regional transmission to occur, and less likely to affect other regions.

### 3.5. Transmission Characteristics of Key Prices

This study selects the top 10 trade prices in terms of influence and conducts an in-depth analysis of their transmission characteristics.

Indonesia’s stainless steel export prices have a strong influence and transmission speed (Figure 7). Indonesia enjoys the advantage of natural nickel resources, creating superior conditions for the rise of its stainless steel industry. In addition, stainless steel companies such as Castle Peak Steel Group have built plants in Indonesia, which has led to a rapid expansion of stainless steel production capacity in Indonesia [52]. Relying on the advantages of industry chain integration, the lowest point of global stainless steel production cost will be shifted from China to Indonesia. The International Stainless Steel Forum forecast data shows that Indonesia overtakes India to become the second-largest stainless steel producer in the world in 2022 [44]. With the expansion of stainless steel demand in emerging economies such as ASEAN and India, these countries and regions will become important destinations for Indonesia’s stainless steel exports. The influence and transmission speed of Indonesia’s stainless steel export prices will further increase.

Ferronickel import prices in Italy, ferronickel export prices in Ukraine, nickel ore import prices in Canada, nickel ore export prices in Albania, and nickel ore import prices in Belgium all have similar characteristics. The impact of these trade prices is not as prominent compared to Indonesia’s stainless steel export prices, but all have a high rate of transmission. The reason is the high concentration of trading partners in these countries and the existence of certain geo-trade characteristics. For example, Italy’s ferronickel imports are mainly to European and American countries such as Guatemala, Ukraine, and Brazil. Likewise, its ferronickel import competition partners are mostly concentrated in these regions. For another example, Italy and the UK are the main import partners of Ukraine ferronickel.

Germany’s ferronickel exports have a strong transmission sensitivity and transmission pivot. Germany’s nickel resources are scarce, so the German economy is based on heavy industry and high-value-added technology exports. Large imports of nickel resources are a prerequisite for the normal production of its nickel-related industries. In particular, nickel ore trade prices are subject to high volatility risks. When the upstream raw material market changes, Germany’s ferronickel export prices will be severely impacted. Germany’s nickel ore import prices have similar characteristics. At the same time, Germany’s nickel ore import prices have a high transmission rate due to the fact that the US is the main source of its nickel ore imports.

Russian nickel ore export prices have a high transmission speed and transmission pivot. Russia is an important supplier of nickel metal consumption to European countries, such as the Russian Norilsk Nickel Group, which is one of the largest nickel producers in the world [53]. After the Russia-Ukraine conflict, many EU countries stopped trading with Russia, as well as cut off some sea transport channels, which will directly lead to an increase in the cost of Russia’s nickel ore exports. It can be expected that the fluctuation of Russia’s nickel ore export price will quickly affect the price of nickel products in European countries, and even have an impact on the transmission between the prices of other nickel products.

## 4. Conclusions

The main objective of this study is to use systematic risk entropy and weighted directed Granger causal networks. The volatility risk of trade prices of nickel products is measured from an industry chain perspective and the transmission patterns of different volatility risk prices are explored. The results of the study are important for stabilizing trade prices, ensuring the security of the supply of nickel resources, and establishing a good international trade market. The results of the study are as follows. 

(1)Nickel ore trade prices have the highest exposure to volatility, with the highest exposure to volatility in import prices. Trade prices for stainless steel downstream of the chain are mainly influenced by end demand and are more stable than nickel ore prices. The entropy of risk is greater than 0.9 for all regions, except the Americas, where there is a high risk of volatility in trade prices and a low risk of price volatility.(2)There is a positive relationship between the impact, sensitivity degree, and transmission hub of nickel trade prices, while there is no significant relationship between the transmission speed and the other indicators. The main transmission characteristics of trade prices are low impact, low degree of being influenced, slow transmission speed, and weak transmission hub, which indicates that the Granger causal network composed of trade prices has obvious scale-free characteristics.(3)The trade price of stainless steel has a higher influence and is also more susceptible to the influence of other trade prices. Stronger bargaining power for stainless steel exporters. The export price of ferronickel is more susceptible to other trade prices. The risk of high volatility in nickel ore prices is not easily passed on to other parts of the chain. Nickel ore trade prices have a strong influence on the transmission between other prices, so the higher volatility risk of nickel ore trade prices still needs to be watched. Upstream and midstream in the chain are more likely to have a price transmission relationship with downstream. There is a longer transmission chain between upstream prices and downstream prices.(4)There is a strong influence and sensitivity to trade prices in Asia and Europe. Asia’s trade price influence is mainly derived from stainless steel prices, while Europe’s price influence is concentrated in the upstream and midstream of the chain. Compared to Asia and Europe, other regions do not have as much trade price transmission capacity. Asia is more likely to have a price-transmission relationship with Europe, and there is a longer transmission chain between Asia and Europe. Many price transmission paths also exist within continents, with Europe and Asia in particular predominating. When their prices change, they are prone to both intra- and inter-regional transmission, and this transmission does not usually affect other regions.(5)Indonesia’s stainless steel export prices have a strong impact and transmission speed. Germany’s ferronickel exports have a strong transmission sensitivity and transmission hub. Russian nickel ore export prices have a high transmission speed and transmission hub. Resource advantages, foreign investment, industry chain integration, and geo-country demand expansion will further increase the influence and speed of transmission of Indonesian stainless steel export prices. A higher concentration of trading partners and geo-trade characteristics can increase the price transmission speed. Imports of nickel resources underpin the development of heavy industry in Germany and the export of high-value-added technologies. Upstream price fluctuations affect the export prices of German midstream products. Russia is an important supplier of nickel metal consumption to European countries. Fluctuations in the price of Russia’s nickel ore exports will quickly affect the price of nickel products in European countries and will even have an impact on the price transmission between other nickel products.

Based on the results of the study, we make the following recommendations:(1)As the risk of volatility in the price of nickel ore trade is the highest, it is recommended that governments pay attention to their nickel stocks and set a safe level of nickel stocks based on their demand, to cope with the possible resource supply crisis brought about by sharp fluctuations in upstream prices and increase stocks to cope with the risk.(2)The trade price of stainless steel has a stronger influence. The bargaining power of stainless-steel exporters is stronger than that of importers. Therefore, Brazil, New Caledonia, and other resource countries should absorb foreign investment to form a more complete integrated industrial chain, improve the smelting and processing capacity of nickel ore, and transform the resource advantage into economic advantage.(3)The international trade ties are getting closer, Russia is an important supplier of resources to Europe, and international trade sanctions will have a negative impact on countries. Countries should take the initiative to maintain mutually beneficial trade relations and establish a healthy and good trade market.

## Figures and Tables

**Figure 1 entropy-24-01221-f001:**
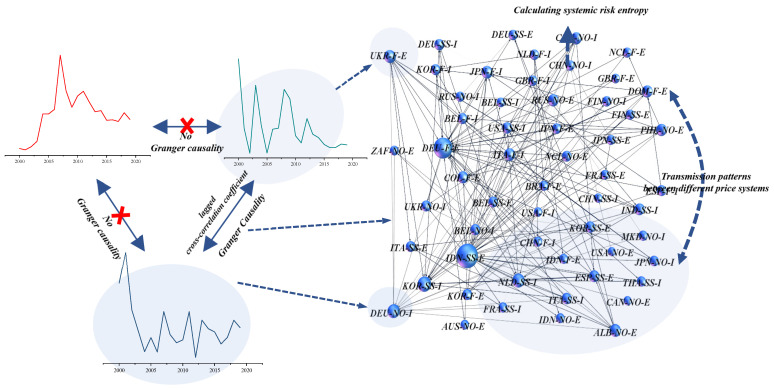
Granger causal network based on the price of nickel products.

**Figure 2 entropy-24-01221-f002:**
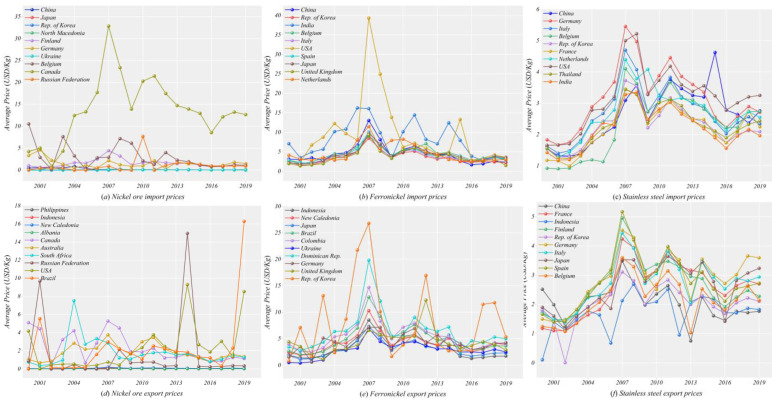
Fluctuating trends in import and export prices of nickel products from 2000 to 2019 (top 10 countries by price). Where (**a**–**c**) is the import price and (**d**–**f**) is the export price.

**Figure 3 entropy-24-01221-f003:**
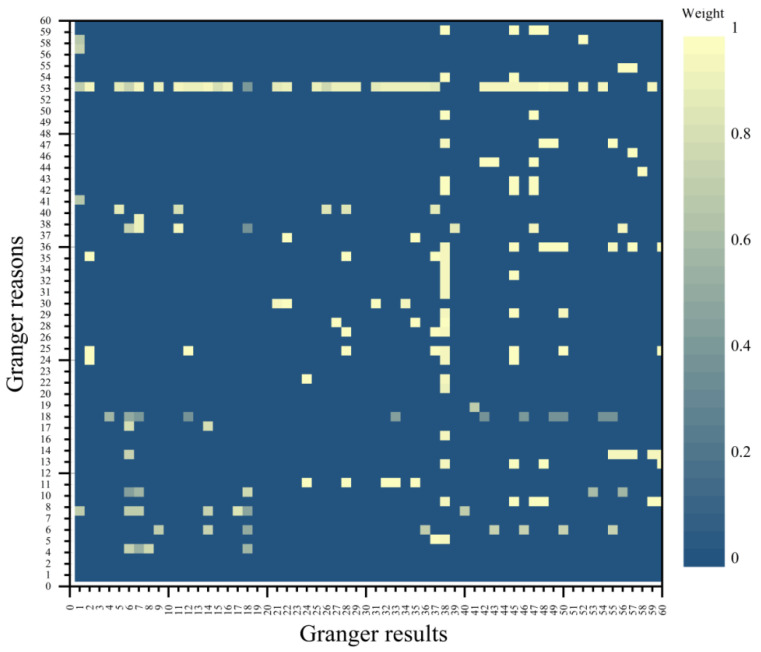
The adjacency matrix heat map. (The vertical axis represents Granger cause, and the horizontal axis represents Granger result, that is, the influence of the vertical axis trade price on the horizontal axis trade price. The depth of the color block represents the strength of the causal relationship between trade prices, that is, the brighter the color, the stronger the causal relationship).

**Figure 4 entropy-24-01221-f004:**
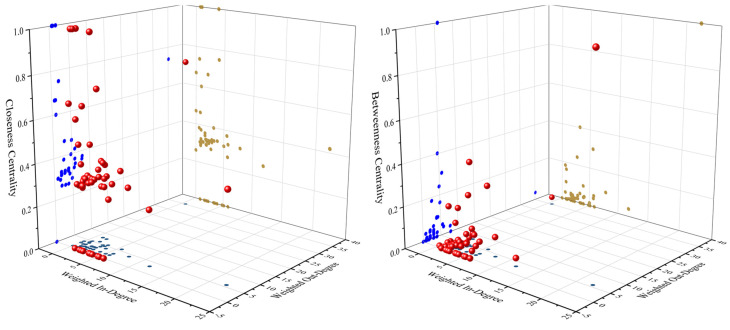
Interrelationships among indicators and the characteristics of the distribution in nickel product prices (dots with different colors show projections in different directions).

**Figure 5 entropy-24-01221-f005:**
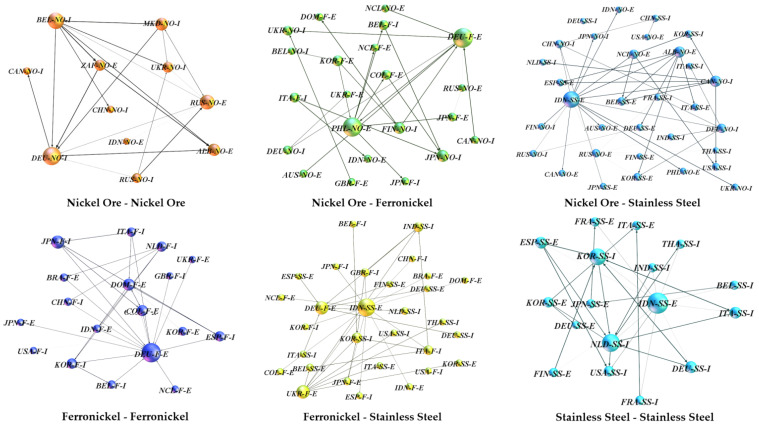
Diagram of the price transmission network among different nickel products (The weighted degree of trade price determines the size of the node. The Granger causality between prices determines the edge thickness).

**Figure 6 entropy-24-01221-f006:**
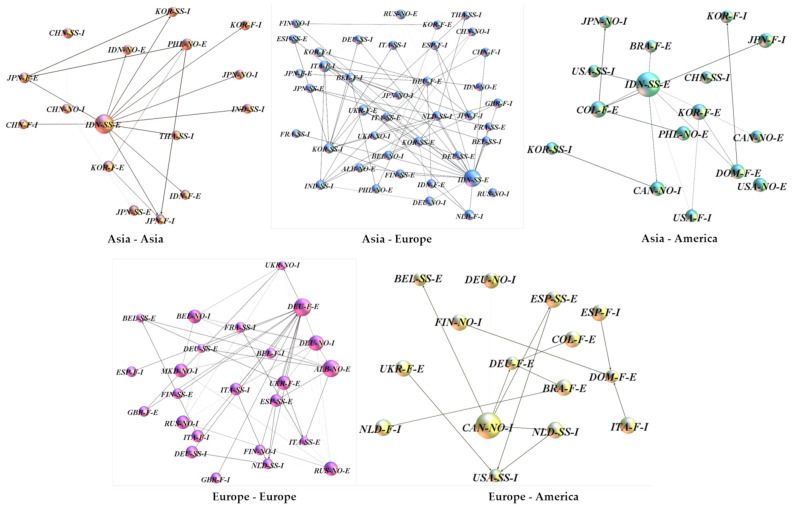
Diagram of the price transmission network among different regions. (The weighted degree of trade price determines the size of the node, and the Granger causality between prices determines the edge thickness).

**Figure 7 entropy-24-01221-f007:**
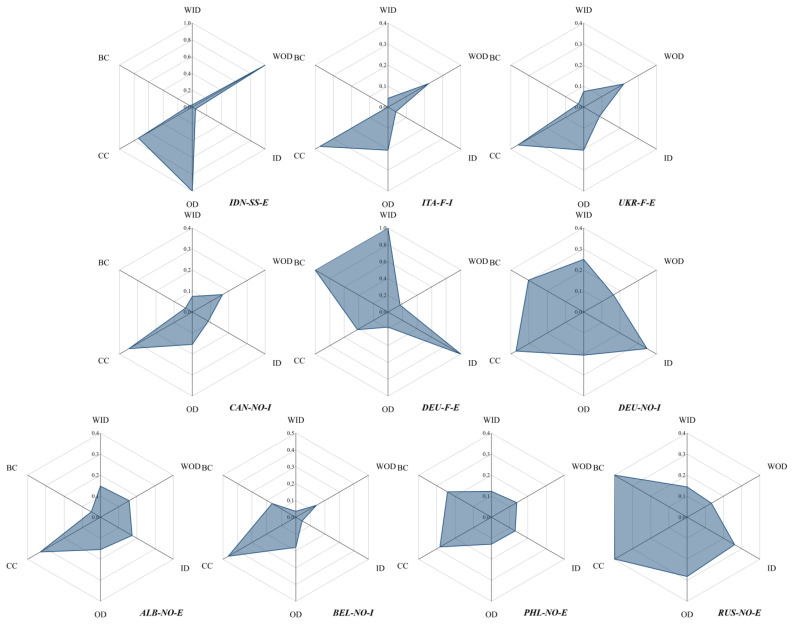
Transmission characteristics of the top 10 nickel product prices by impact.

**Table 1 entropy-24-01221-t001:** Risk of fluctuations in import and export prices of nickel ore, ferronickel, and stainless steel.

Nickel Ore Import Prices	Nickel Ore Export Prices	Ferronickel Import Prices	Ferronickel Export Prices	Stainless Steel Import Prices	Stainless Steel Export Prices
0.8317	0.8909	0.9325	0.9272	0.939	0.9367

**Table 2 entropy-24-01221-t002:** Price fluctuation risk of Nickel product in Asia, Europe, America, and Oceania.

Asia	Europe	America	Oceania
0.9326	0.9258	0.8543	0.912

**Table 3 entropy-24-01221-t003:** Transmission characteristics of trade prices for different products.

Product	Nickel Ore		Ferronickel		Stainless Steel	
Trade Type	Import	Export	Import	Export	Import	Export
WOD	2.685	2.477	2.924	2.941	1.746	5.051
	2.581		2.933		3.311	
WID	2.538	1.661	1.980	4.286	4.263	2.668
	2.099		3.194		3.507	
OD	3.556	3.333	3.000	3.200	1.800	5.556
	3.444		3.105		3.579	
ID	3.333	2.222	2.111	4.600	4.700	3.000
	2.778		3.421		3.895	
CC	0.262	0.288	0.283	0.390	0.381	0.459
	0.275		0.339		0.418	
BC	0.078	0.077	0.015	0.130	0.020	0.015
	0.077		0.076		0.018	

**Table 4 entropy-24-01221-t004:** Parameters of the price transmission network among different nickel products.

Type of Transmission	Nodes	Edges	Diameter	Average Path Length
Nickel ore—Nickel ore	11	23	5	2.125
Nickel ore—Ferronickel	22	24	5	2.324
Nickel ore—Stainless steel	32	42	3	1.655
Ferronickel—Ferronickel	19	32	4	1.708
Ferronickel—Stainless steel	30	36	4	1.621
Stainless steel—Stainless steel	17	32	4	1.813

**Table 5 entropy-24-01221-t005:** Transmission characteristics of trade prices for different products in Asia and Europe.

Region	Asia			Europe		
Products	Nickel Ore	Ferronickel	Stainless Steel	Nickel Ore	Ferronickel	Stainless Steel
WOD	1.222	2.162	6.345	3.569	4.368	1.682
	3.663			3.037		
WID	3.107	2.563	3.779	2.529	3.796	3.119
	3.192			3.145		
OD	1.250	2.333	7.000	5.375	4.625	1.727
	4.000			3.667		
ID	3.750	2.833	4.286	3.625	4.000	3.455
	3.647			3.667		
CC	0.071	0.311	0.581	0.343	0.422	0.353
	0.366			0.371		
BC	0.060	0.037	0.019	0.139	0.133	0.019
	0.035			0.088		

**Table 6 entropy-24-01221-t006:** Transmission characteristics of trade prices for different products in America and Oceania.

Region	America			Oceania		
Products	Nickel Ore	Ferronickel	Stainless Steel	Nickel Ore	Ferronickel	Stainless Steel
WOD	2.184	1.711	0.000	2.423	0.963	-
	1.675			1.936		
WID	0.819	3.258	5.875	0.913	1.903	-
	2.670			1.243		
OD	2.333	1.750	0.000	2.500	1.000	-
	1.750			2.000		
ID	1.000	3.500	6.000	1.000	2.000	-
	2.875			1.333		
CC	0.338	0.224	0.000	0.313	0.300	-
	0.239			0.309		
BC	0.013	0.037	0.000	0.000	0.006	-
	0.023			0.002		

**Table 7 entropy-24-01221-t007:** Parameters of the price transmission network among different regions.

Type of Transmission	Nodes	Edges	Diameter	Average Path Length
Asia—Asia	16	19	3	1.182
Asia—Europe	40	71	9	3.14
Asia—America	16	16	2	1.333
Europe—Europe	24	53	4	2.436
Europe—America	15	14	3	1.381

## Data Availability

Not applicable.

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
