# Peer review of "Risk Transmission of Trade Price Fluctuations from a Nickel Chain Perspective: Based on Systematic Risk Entropy and Granger Causality Networks"

_entropy, 2022, doi:10.3390/e24091221_

Round 1

Reviewer 1 Report

The subject of the article is timely and is interesting for research. 

In general, the article has an acceptable structure, but some parts need improvement.

The abstract is too long (it must have a maximum of 200 words) and is not well structured. It should be reviewed and rewritten in a more concise manner.

When ne reference has more than one author, it should be mentioned in the text (see for example Xiao (2018)).

I don't think it's necessary to use subtitles 2.2.1-2.2.4, 2.3.1.-2.3.6.

Some figures are difficult to follow.

Also, the quality of English can be improved.

Author Response

Dear Reviewer, We have made new changes to the manuscript as you requested. Please see the attached document for details of the changes.

Reviewer 2 Report

The methods should be more concretely described with appropriate references, procedures of testing, statistical inference theories in support of the method, etc.

Author Response

(The authors gave the same response as above.)

Reviewer 3 Report

This article considers risk transmission of nickel prices based on systemic risk entropy and Granger causality networks. Authors extract network from the combination of Granger causality and time lagged correlation coefficients. They introduced transmission speed and systemic risk entropy. I have some questions and comments. Authors should be revised the article following the comments.

Major Comments:

1.     Authors defined transmission speed in Eq. (12). This definition is closeness centrality in complex network theory. It is very curious that authors call this centrality as a transmission speed. It is the average distance form the focal node i to the rest of the network. Give some comments why Eq. (12) is called transmission speed in the causality network.

2.     In the definition of systemic risk entropy, authors used the eigenvalue of the matrix WDGCN. The form of entropy is like to Shannon entropy if lambda_i/N is a probability. However, lambda_i/N is not a probability. Therefore, it is hard to call it as entropy. It is uncertain that Eq. (14) is measuring the systemic risk. Discuss these points.

Minor comments

159: ADF à Give full terminology

161: Where à where,  

     Identify what y_t is.  Y_t is the price of nicke ore…

172: Where à where

195: GCji is 1?

223: proximity centrality à closeness centrality

234: eq (13) is identical to eq (12). Check eq. (13).

241: It is uncertain eq. (14) is measuring volatility risk.

Fig.1, Fig. 5 à Increase line thickness

388: in à In

Author Response

Dear reviewers, we have carefully taken into account your suggestions and have revised the manuscript. Please see the attached document for details of the revisions.

Round 2

Reviewer 3 Report

The manuscript revised according to reviewer's comments and the quality of the article is improved. I accept to publish.